# Molecular Detection of *Candidatus* Rickettsia andeanae and *Ehrlichia* sp. in *Amblyomma pseudoconcolor* Aragão, 1908 (Acari: Ixodidae) from the Argentinian Patagonia

**DOI:** 10.3390/ani12233307

**Published:** 2022-11-26

**Authors:** Patrick Stephan Sebastian, Marina Winter, Sergio Damián Abate, Evelina Luisa Tarragona, Santiago Nava

**Affiliations:** 1Instituto de Investigación de la Cadena Láctea (IdICaL), CONICET—INTA, Ruta 34 km 227, 2300 Rafaela, Santa Fe, Argentina; 2Centro de Investigaciones y Transferencia de Río Negro (CONICET-UNRN), Universidad Nacional de Río Negro, Sede Atlántica, Avenida Don Bosco 500, 8500 Viedma, Río Negro, Argentina

**Keywords:** Rickettsiales, *Amblyomma pseudoconcolor*, ticks, *Chaetophractus villosus*, wildlife, Argentina

## Abstract

**Simple Summary:**

The study presents the molecular detection of two bacterial agents in a hard tick (*Amblyomma pseudoconcolor*) collected on a large hairy armadillo (*Chaetophractus villosus*) from the Argentinian Patagonia. Molecular detection of bacterial agents was performed by polymerase chain reaction (PCR). One tick, determined morphologically and genetically as *A. pseudoconcolor*, was collected on *C. villosus*. The bacterial agents detected in the hard tick were identified as *Candidatus* Rickettsia andeanae and *Ehrlichia* sp. The results of this study and previous findings suggest that *A. pseudoconcolor* may be a potential vector of some *Rickettsia* and *Ehrlichia* bacteria of unknown pathogenicity.

**Abstract:**

This study presents the molecular detection of *Candidatus* Rickettsia andeanae and *Ehrlichia* sp. in *Amblyomma pseudoconcolor* Aragão, 1908 (Acari: Ixodidae) collected on a large hairy armadillo (*Chaetophractus villosus* (Desmarest, 1804)). On 12 October 2020, a specimen of *C. villosus* was found dead on the road in Río Negro province, Argentina. Molecular detection of *Rickettsia* and *Ehrlichia* agents was performed amplifying the *gltA* and 16S rRNA gene, respectively. One tick, determined morphologically and genetically as *A. pseudoconcolor*, was collected on *C. villosus*. The rickettsial agent detected in *A. pseudoconcolor* was identified as *Candidatus* Rickettsia andeanae. The *Ehrlichia* sp. strain showed high sequence similarity to different uncultured *Ehrlichia* sp. detected in horses, capybaras and *Ixodes ornithorhynchi* from Nicaragua, Brazil and Australia, respectively. The results of this study and previous findings suggest that *A. pseudoconcolor* may be a potential vector of some *Rickettsia* and *Ehrlichia* bacteria of unknown pathogenicity.

## 1. Introduction

The large hairy armadillo, *Chaetophractus villosus* (Desmarest, 1804) (Cingulata: Chlamyphoridae) is one of the largest species of the order Cingulata in South America and the most abundant species of armadillos in Argentina [1]. The geographical distribution of this mammal reaches from the southeast of Bolivia and west of Paraguay to the south of Argentina, including one population situated in the Province of Tierra del Fuego in the extreme south of the Southern Cone of South America [1]. *Chaetophractus villosus* prefers inhabiting mostly sandy and open soils such as steppes and hills, but also intermountain valleys, plains and grasslands. Further, it could be found in human-modified environments, including peri-urban areas [1,2]. *Chaetophractus villosus* is frequently parasitized by ticks from the genus *Amblyomma*, especially the species *Amblyomma auricularium* (Conil, 1878) and *Amblyomma pseudoconcolor* Aragão, 1908 [3,4].

*Amblyomma pseudoconcolor* Aragão, 1908 (Acari: Ixodidae) is an endemic tick species in Argentina that is distributed from the Neotropical Region to the Andean Region of the Southern Cone of America, including the biogeographic provinces as defined by Morrone [5]: Chaco, Pampa, Monte and Central Patagonia, from the north to the south [3,6]. Its main hosts for all life stages (larvae, nymphs, adults) are mammals of the family Dasypodidae and Chlamyphoridae [4]. Parasitism of *A. pseudoconcolor* on *C. villosus* from the Argentinean Patagonia was described by Ezquiaga et al. [7]. *Amblyomma pseudoconcolor* is often confused morphologically with *A. auricularium* that also parasites on armadillos (Cingulata) (Nava et al., 2017; Guglielmone et al., 2021) [3,4]. However, *A. pseudoconcolor* presents a typical scutal ornamentation, which is not seen in *A. auricularium* [4]. Three different rickettsial agents of unknown pathogenicity were detected in *A. pseudoconcolor*: *Candidatus* Rickettsia andeanae [8,9], *Rickettsia amblyommatis* [10], and *Rickettsia bellii* [11]. In addition, human parasitism of *A. pseudoconcolor* is described in one case [8].

The aim of this study was to detect and identify the possible presence of bacteria belonging to the order Rickettsiales in a female specimen of *A. pseudoconcolor* collected on *C. villosus* from Río Negro province, Patagonia, Argentina.

## 2. Materials and Methods

On 12 October 2020, a male adult specimen of *C. villosus* was found dead on the road Ruta Provinicial Nº1 between the locations of Viedma and Balneario El Cóndor (40°56′44.88″ S; 62°51′40.75″ W), Río Negro, Argentina. The animal was examined manually for the presence of ticks. The tick collected free on the animal was sent to the Instituto Nacional de Tecnología Agropecuaria Rafaela, Santa Fe, Argentina, for identification and detection of tick-borne bacteria. Firstly, the tick was identified morphologically according to Nava et al. [3]. Afterwards, complete DNA was extracted using the High Pure PCR Template Preparation Kit (Roche, Mannheim, Germany) according to the manufacturer’s instructions. To confirm the morphological identification, a PCR detecting a specific fragment of the mitochondrial 16S rRNA gene of members of the order Ixodidae was processed using the primers 16S + 1 (5′-CCG GTC TGA ACT CAG ATC AAG T-3′; [12]) and 16S-1 (5′-GCT CAA TGA TTT TTT AAA TTG CTG T-3′; [12]) following Mangold et al. [13]. To detect bacteria from the order Rickettsiales, three different conventional PCR assays were used: (I) Amplification of a 345 bp fragment of the 16S rRNA gene of the family Anaplasmataceae (used primers: GE2-F2 (5′- GTT AGT GGC AGA CGG GTG AGT-3′) and HE3 (5′-TAT AGG TAC CGT CAT TAT CTT CCC TAT-3′) [14,15,16], (II) amplifying an 830 bp fragment of the *Rickettsia* genus specific gene for a citrate synthase—*gltA* (primers: CS-239: 5′-GCT CTT CTC ATC CTA TGG CTA TTA T-3′; CS-1069: 5′-CAG GGT CTT CGT GCA TTT CTT) [17], and (III) amplification of an approximate 530 bp fragment of the gene for a 190-kDa outer membrane protein (*ompA*) that is specific for *Rickettsia* sp. of the spotted fever group Rickettsiae (primer names and sequences: Rr 190.70p; 5′-ATG GCG AAT ATT TCT CCA AAA-3′ and Rr 190.602n; 5′-AGT GCA GCA TTC GCT CCC CCT-3′) [18]. In all PCR reactions, ultra-pure water was used as negative control while DNA of *Ehrlichia canis* and *Rickettsia massiliae* acted as positive control for the detection of the genes for 16S rRNA and gltA, respectively. Positive PCR amplicons of the three assays were purified using the High Pure PCR Product Purification Kit (Roche, Mannheim, Germany) and sent to INTA Castelar (Genomics Unit, Buenos Aires, Argentina) for sequencing. Obtained partial sequences were edited using BioEdit Sequence Alignment Editor [19] with manual edition whenever it was necessary, aligned with the program Clustal W [20], and compared with sequences deposited in GenBank. Phylogenetic analyses were performed with maximum-likelihood (ML) methods by using the program Mega X [21]. Best-fitting substitution models were determined with the Akaike Information Criterion using the ML model test implemented in MEGA X. Support for the topologies was tested by bootstrapping over 1.000 replications, and gaps were excluded from the comparisons.

The present work was executed with the permission from the Secretariat of Environment and Climate Change of the Province of Río Negro (File No. 08526SAYDS 2015/218/222).

## 3. Results and Discussion

The female tick that was collected on *C. villosus* was identified morphologically as *A. pseudoconcolor* based on the typical scutum ornate, with small pale spots on the yellowish-brown ground with small, moderately deep punctuations, the dental formula (3/3) and the spurs on the trochanters [3] (see Figure 1). Further, the identification was confirmed by sequencing a fragment of the mitochondrial 16S rRNA gene. The obtained partial sequence (GenBank accession number: OP744428) showed sequence identities of 99.30% and 99.53% to *A. pseudoconcolor* from Argentina (GenBank accession numbers AY628134 and AY628135). The tick specimen was deposited in the tick collection of the Instituto Nacional de Tecnología Agropecuaria (INTA; Rafaela, Santa Fe, Argentina) with the collection number INTA 2521 (see Figure 1).

The tick sample showed positive PCR results in two of the three applied assays: 16S rRNA gene for Anaplasmataceae and *gltA* for *Rickettsia* spp. The two amplicons were purified and partial gene sequences were obtained. The partial sequence of the *gltA* gene (GenBank accession number: OP753007) showed a sequence identity of 100% (653/653bp) to different *Ca*. R. andeanae sequences from isolates made from *Amblyomma maculatum* Koch, 1844, *Amblyomma parvum* Aragão, 1908 and *Amblyomma tigrinum* Koch, 1844 from Argentina, Brazil, Peru and the United States (GenBank accession numbers: EF451001, GU131156, GU169050, KT153033 and KX576677). In the ML tree based on GenBank sequences of different species of *Rickettsia*, the gltA sequence generated in this study forms part of a clade including sequences of *Ca*. R. andeanae detected in different species of *Amblyomma* from the Southern cone of South America (see Figure 2). This clade separates clearly (bootstrap value 97) from other species of *Rickettsia*. *Candidatus* R. andeanae is a member of the spotted fever group Rickettsiae (SFGR) of unknown pathogenicity [9,22,23]. Interestingly, the PCR assay amplifying the *ompA* gene that is specific for SFGR resulted negative in this study. However, previous studies have shown [24,25] that the applied PCR assay in this study is not appropriate for the detection of the *ompA* gene of *Ca.* R. andeanae and therefore must be replaced by another assay described by Ermeeva et al. [26] in future studies. Jiang et al. [27] firstly described the detection of *Ca.* R. andeanae in *A. maculatum* from Peru. Further, this rickettsial agent could be detected in *A. parvum* from Argentina and Brazil [24,25,28] and *A. tigrinum* from Argentina and Chile [29,30]. DNA of *Ca*. R. andeanae (named as *Rickettsia* sp. strain Argentina by the authors) was previously detected in *A. pseudoconcolor* collected in Santiago del Estero province, Argentina [8]. Santiago del Estero province belongs to the Chaco biogeographic province, which is ecologically distinct from the Central Patagonia biogeographic province where *A. pseudoconcolor* was sampled in this study [5]. Further, *Ca*. R. andeanae was also detected in *Ixodes boliviensis* collected from a horse in Peru [27]. Nevertheless, the results of the present study together with the reports from literature suggest that this SFGR is closely associated with the hard tick genus *Amblyomma* and widely distributed in South America.

The partial sequence of the 16S rRNA gene from the tick simple that was generated in this study showed a sequence identity of 99.53% (305/307bp) to an uncultured *Ehrlichia* sp. detected in a horse from Nicaragua (GenBank accession number: KJ434178) and 98.69% to uncultured *Ehrlichia* spp. from *Ixodes ornithorhynchi* Lucas, 1846 (Australia; 301/305bp; GenBank accession number: MF069159) and *Hydrochoerus hydrochaeris* Linnaeus, 1766 (Brazil; 301/305bp; GenBank accession number: MW785880). Figure 3 shows the phylogenetic tree constructed of partial 16S rRNA gene sequences of different *Ehrlichia* spp. Based on the used fragment of the 16S rRNA gene, the phylogenetic position of the *Ehrlichia* sp. detected in this study (GenBank accession number: OP744461) in relation to other *Ehrlichia* spp. remains unresolved. In Argentina, five species of *Amblyomma* have been reported infected with different strains of undetermined *Ehrlichia* sp. so far.: *Amblyomma neumanni* Ribaga, 1902 [31]; *Amblyomma ovale* Koch, 1844 [32]; *A. parvum* [33,34]; *A. tigrinum* [33,34,35,36]; and *Amblyomma triste* Koch, 1844 [37]. The *Ehrlichia* strain detected in *A. pseudoconcolor* in this work is not related to the remaining *Ehrlichia* strains previously detected in Argentina, at least considering those from which 16S rRNA gene sequences are available. Based on the results of this studies together with previous reports, it must be assumed that the diversity of *Ehrlichia* spp. in *Amblyomma* ticks from Argentina is greater than previously suggested. However, further studies using genetic markers with a higher level of polymorphism should be performed for a more accurate phylogenetic characterization of these *Ehrlichia* spp. In addition, the *Ehrlichia*–host relationship and the possible pathogenicity of these strains must be studied more in detail.

## 4. Conclusions

The results of this study demonstrate the presence of a spotted fever group *Rickettsia*—*Ca.* R. andeanae—and the first detection of a putative novel strain of *Ehrlichia* sp. in *A. pseudoconcolor.* Further studies must be executed to investigate if *A. pseudoconcolor* may act as a vector for these bacteria and which is the role of the *C. villosus* in this bacteria–tick–host relationship.

## Figures and Tables

**Figure 1 animals-12-03307-f001:**
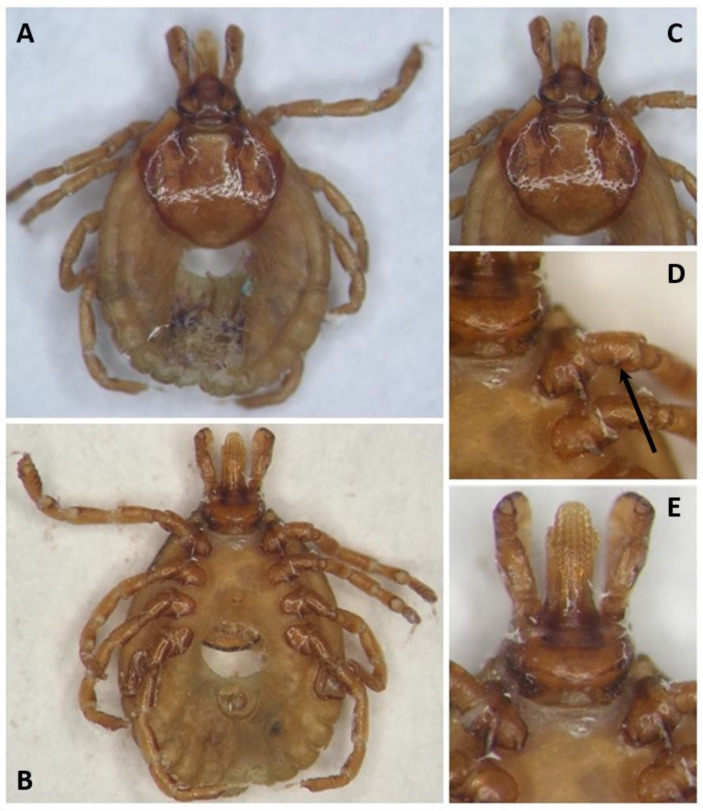
*Amblyomma pseudoconcolor* female (Tick collection Instituto Nacional de Tecnología Agropecuaria (INTA; Rafaela, Santa Fe, Argentina); collection number INTA 2521). (**A**) dorsal view; (**B**) ventral view; (**C**) scutum ornamentation; (**D**) coxae and trochanters; (**E**) capitulum ventral view. The black arrow in (**D**) indicates the typical spur on the trochanter.

**Figure 2 animals-12-03307-f002:**
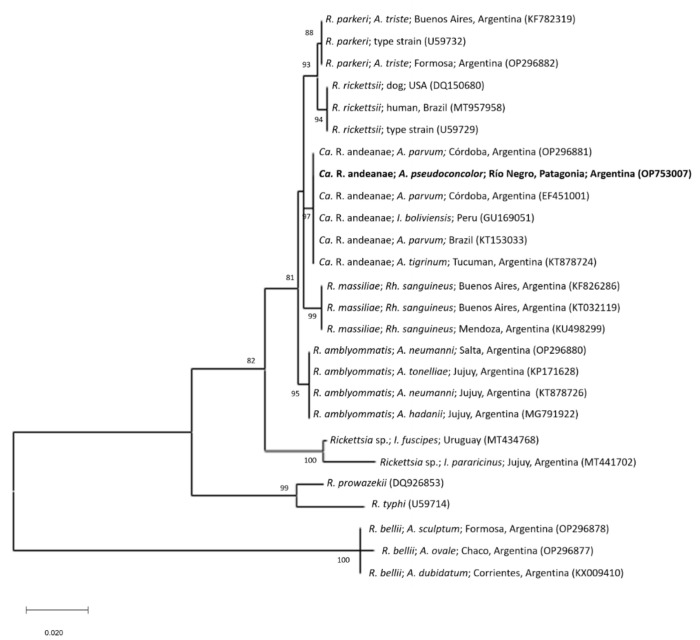
Maximum-likelihood tree constructed from gltA partial sequences of different *Rickettsia* species (Substitution model: Tamura-Nei 93 model with Gamma distribution). The sequence generated in this study is written in bold letters. Numbers represent bootstrap support generated from 1000 replications. GenBank accession numbers are given in brackets. Abbreviations: *A*.: *Amblyomma*; *An*.: *Anaplasma*; *Ca*.: *Candidatus*; *I*.: *Ixodes*; *R*.: *Rickettsia*; *Rh*.: *Rhipicephalus*.

**Figure 3 animals-12-03307-f003:**
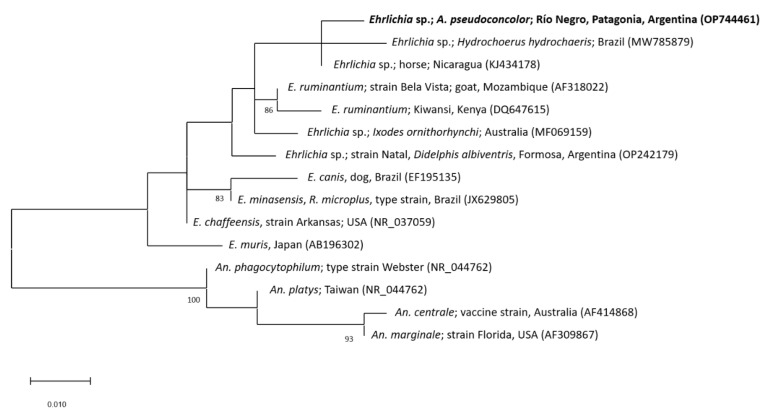
Maximum-likelihood tree constructed from 16S rRNA gene partial sequences of different *Ehrlichia* species (Substitution model: Hasegawa–Kishino–Yano model with Gamma distribution). Sequences of *Anaplasma* spp. were used as outgroup. The sequence generated in this study is written in bold letters. Numbers represent bootstrap support generated from 1000 replications. Bootstrap values minor to 80 are not shown. GenBank accession numbers are given in brackets. Abbreviations: *A*.: *Amblyomma*; *An*.: *Anaplasma*; *E*.: *Ehrlichia*; *R*.: *Rhipicephalus*.

## Data Availability

Not applicable.

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
