# Peer review of "Molecular Detection of *Candidatus* Rickettsia andeanae and *Ehrlichia* sp. in *Amblyomma pseudoconcolor* Aragão, 1908 (Acari: Ixodidae) from the Argentinian Patagonia"

_animals, 2022, doi:10.3390/ani12233307_

Round 1
Reviewer 1 Report
The manuscript is very well written and easy to read. No major comments on this regards.
Minor comments:
Conclusions section: It is Ok to speculate that this Ehrilchia sp. may represent “a putative novel strain of Ehrlichia sp.”. Nevertheless, a 99.53% (305/307bp) homology in the 16SrRNA may be well represent a mutations associated with a different strain and not as a new species. The authors should state that until other genes are sequences, the identification of this as new species is apparent.
Line 32: Italicize “A. pseudoconcolor”
Line 138: Italicize “Ehrlichia sp.”
Author Response
Response to Reviewer #1
The manuscript is very well written and easy to read. No major comments on this regards.
- Thank you for your comments and suggestions.
Minor comments:
Conclusions section: It is Ok to speculate that this Ehrlichia sp. may represent “a putative novel strain of Ehrlichia sp.”. Nevertheless, a 99.53% (305/307bp) homology in the 16SrRNA may be well represent a mutations associated with a different strain and not as a new species. The authors should state that until other genes are sequences, the identification of this as new species is apparent.
- Topic is mentioned in lines 162-164.
Line 32: Italicize “A. pseudoconcolor”
- Changed in the text (line 32).
Line 138: Italicize “Ehrlichia sp.”
- Changed in the text (lines 148, 153).
Reviewer 2 Report
Dear authors
The manuscript “Molecular detection of Candidatus Rickettsia andeanae and Ehrlichia sp. in Amblyomma pseudoconcolor Aragão, 1908 (Acari: Ixodidae) from the Argentinian Patagonia” is a well written short communication reporting the presence of two pathogens in this tick species: although the Candidatus Rickettsia andeanae was previously identified in this tick, this study provides the first citation of Ehrlichia in A. pseudoconcolor. I think that this finding is worth to be published. Nevertheless, there are some considerations and suggestions:
-Italics should be used in lines 32, 125, 126, 133 and 138
-Lines 58-61: I think it is worth noting that data on the pathogens found in A. pseudoconcolor is limited. In fact, authors stated that only two different rickettsial agents of unknown pathogenicity were detected in A. pseudoconcolor: Candidatus Rickettsia andeanae and Rickettsia amblyommatis. In addition, the report of Rickettsia bellii in A. pseudoconcolor from Brazil must be included (Costa et al., 2017. New records of Rickettsia bellii-infected ticks in Brazil. Brazilian Journal of Veterinary Research and Animal Science, 54 (1), pp. 92 – 95).
-Lines 62-64: I consider that the objective is not well defined. “Report the molecular detection of a putative novel strain of Ehrlichia sp. and Ca. R. andeanae in a female specimen of A. pseudoconcolor”? I think that a more suitable objective was to identify the possible presence of pathogens belonging to the Anaplasmataceae family and Rickettsia genus in a female specimen of A. pseudoconcolor collected on C. villosus from Río Negro province, Patagonia, Argentina.
-Line 70: Was the tick found free on the animal? Was it feeding on the animal?
-Lines 78-80: Only one gene was used for identifying both pathogens. I think that sequence analysis at other genes is needed for an accurate identification. In fact, authors stated this in lines 147-149: “further studies using genetic markers with a higher level of polymorphism should be performed for a more accurate phylogenetic characterization of these Ehrlichia spp.” The gltA gene is not very informative for Rickettsia identification. Did you use other genes (such as rOmpA or rOmpB) with negative results? Please, add this information.
-Could you get samples (blood, spleen..) from the large hairy armadillo? It should be interesting to assess if the animal was positive to the same pathogens found in the tick. That could be useful to know if the ticks were positive to the pathogens before feeding (in the case that it was feeding, that it is the most probable).
-Lines 102 and 110: GenBank accession numbers can be obtained in a 2-5 day period. Thus, “GenBank accession number: submitted)” is not acceptable.
-Figure 3: ruminantium should be in lower case.
-Line 171: You do not “demonstrate the detection”. Consider rewriting “demonstrate the presence”. I think it is worth noting to state that this is the first detection of a novel Ehrlichia sp. in A. pseudoconcolor.
-Line 178: Who is X.X.? There are evidences that this work was written quickly…
Author Response
Response to Reviewer #2
Dear authors
The manuscript “Molecular detection of Candidatus Rickettsia andeanae and Ehrlichia sp. in Amblyomma pseudoconcolor Aragão, 1908 (Acari: Ixodidae) from the Argentinian Patagonia” is a well written short communication reporting the presence of two pathogens in this tick species: although the Candidatus Rickettsia andeanae was previously identified in this tick, this study provides the first citation of Ehrlichia in A. pseudoconcolor. I think that this finding is worth to be published. Nevertheless, there are some considerations and suggestions:
Thank you for your comments and suggestions.
-Italics should be used in lines 32, 125, 126, 133 and 138
Changed in the text (lines 32, 121, 134, 148, 153).
-Lines 58-61: I think it is worth noting that data on the pathogens found in A. pseudoconcolor is limited. In fact, authors stated that only two different rickettsial agents of unknown pathogenicity were detected in A. pseudoconcolor: Candidatus Rickettsia andeanae and Rickettsia amblyommatis. In addition, the report of Rickettsia bellii in A. pseudoconcolor from Brazil must be included (Costa et al., 2017. New records of Rickettsia bellii-infected ticks in Brazil. Brazilian Journal of Veterinary Research and Animal Science, 54 (1), pp. 92 – 95).
Sentence was rephrased and information was added (lines 58-61)
-Lines 62-64: I consider that the objective is not well defined. “Report the molecular detection of a putative novel strain of Ehrlichia sp. and Ca. R. andeanae in a female specimen of A. pseudoconcolor”? I think that a more suitable objective was to identify the possible presence of pathogens belonging to the Anaplasmataceae family and Rickettsia genus in a female specimen of A. pseudoconcolor collected on C. villosus from Río Negro province, Patagonia, Argentina.
Objective was changed in the text (lines 62-64).
-Line 70: Was the tick found free on the animal? Was it feeding on the animal?
The tick was found free on the animal. Information was added in the text (line 72).
-Lines 78-80: Only one gene was used for identifying both pathogens. I think that sequence analysis at other genes is needed for an accurate identification. In fact, authors stated this in lines 147-149: “further studies using genetic markers with a higher level of polymorphism should be performed for a more accurate phylogenetic characterization of these Ehrlichia spp.” The gltA gene is not very informative for Rickettsia identification. Did you use other genes (such as rOmpA or rOmpB) with negative results? Please, add this information.
Thank you for this comment. Information was added in the text (lines: 87-90 and lines: 129-133).
-Could you get samples (blood, spleen...) from the large hairy armadillo? It should be interesting to assess if the animal was positive to the same pathogens found in the tick. That could be useful to know if the ticks were positive to the pathogens before feeding (in the case that it was feeding, that it is the most probable).
Unfortunately, no blood or tissue samples from the armadillo were tested to the presence of tick-borne pathogens.
-Lines 102 and 110: GenBank accession numbers can be obtained in a 2-5 days’ period. Thus, “GenBank accession number: submitted)” is not acceptable.
GenBank accession numbers were added to the manuscript and figures (lines: 112, 120 and 153; Figure 2 and 3).
-Figure 3: ruminantium should be in lower case.
Changed in figure 3.
-Line 171: You do not “demonstrate the detection”. Consider rewriting “demonstrate the presence”. I think it is worth noting to state that this is the first detection of a novel Ehrlichia sp. in A. pseudoconcolor.
Changed in the text (lines:186-188)
-Line 178: Who is X.X.? There are evidences that this work was written quickly…
Changed in the text (lines: 191-195)
Round 2
Reviewer 2 Report
Dear authors
All my suggestions and comments have been considered, and I think that this manuscript can be now be published.